# Pushing the thinness limit of silver films for flexible optoelectronic devices via ion-beam thinning-back process

Dongxu Ma[1], Ming Ji[2], Hongbo Yi[2], Qingyu Wang[1], Fu Fan[1], Bo Feng[1,3], Mengjie Zheng[4], Yiqin Chen [1,3] ✉ & Huigao Duan [1,3] ✉

Reducing the silver film to 10 nm theoretically allows higher transparency but in practice leads to degraded transparency and electrical conductivity because the ultrathin film tends to be discontinuous. Herein, we developed a thinning-back process to address this dilemma, in which silver film is first deposited to a larger thickness with high continuity and then thinned back to a reduced thickness with an ultrasmooth surface, both implemented by a flood ion beam. Contributed by the shallow implantation of silver atoms into the substrate during deposition, the thinness of silver films down to 4.5 nm can be obtained, thinner than ever before. The atomic-level surface smooth permits excellent visible transparency, electrical conductivity, and the lowest haze among all existing transparent conductors. Moreover, the ultrathin silver film exhibits the unique robustness of mechanical flexibility. Therefore, the ion-beam thinning-back process presents a promising solution towards the excellent transparent conductor for flexible optoelectronic devices.

Transparent conductive films (TCFs) are indispensable materials in modern optoelectronics[1,2]. To date, transparent conductive oxides (TCOs) (e.g., indium tin oxide (ITO), aluminium-doped zinc oxide (AZO), and indium gallium zinc oxide (IGZO)) are the most used TCFs in commercialized optoelectronic devices[3–5]. However, TCOs suffer from poor mechanical flexibility and insufficient conductance for large-area devices, and there is an increased demand for flexible optoelectronics in the promising wearable device market[6–13]. To address this challenge, many potential alternative TCFs have been proposed, including graphene, carbon nanotubes, silver (Ag) nanowires, and metal meshes. Among them, carbon-based materials can improve mechanical flexibility, but their low electrical conductance greatly limits the electrical performance[14]. Ag nanowires and metal meshes can achieve high transparency with a low sheet resistance, but their fabrication requires either patterning or complicated wet-chemical processes. Meanwhile, both nanowire and metal mesh platforms only provide global conductivity, in which the surface are non-conductive between the wires[15–17]. Therefore, they cannot be used as transparent conductors for those optoelectronic devices which require continuous areal conductance.

A thin and continuous metal film is another promising alternative solution as TCFs. Ag film and its derivatives are supposed to be the most appropriate choices because Ag has the highest electrical conductance and the lowest optical loss in the visible region among all metals[18]. To achieve the highest transparency, producing a large-area sufficiently thin continuous film is required. However, ultrathin Ag films (UTAFs) less than 10 nm tend to be porous or island-shaped according to the Volmer-Weber model[8], which significantly degrades the transparency and film conductivity due to unwanted photon and electron scattering. To suppress island-like growth, improving the wettability of Ag on substrates is a direct method[19]. Introducing a high-wettability metal layer (e.g., Al, Ti, Cu) between the Ag film and substrate or doping such metals in the Ag film is commonly used[20–24]. However, these metals have a high optical loss, which degrades the transparency of the Ag film. Dielectric wetting layers are helpful to avoid the optical loss of the above metal wetting

[1]College of Mechanical and Vehicle Engineering, Hunan University, Changsha, Hunan Province, China. [2]IBD Technology Co., Ltd., Zhongshan, Guangdong Province, China. [3]Greater Bay Area Institute for Innovation, Hunan University, Guangzhou, Guangdong Province, China. [4]Jihua Laboratory, Foshan, Guangdong Province, China. ✉e-mail: chenyiqin@hnu.edu.cn; duanhg@hnu.edu.cn

layers, but the ultimate thinness of Ag films is still limited because complete wetting of Ag adatoms is difficult due to the greatly larger surface energy of Ag metal compared to its adhesion energy with the dielectric substrates[25–28]. The above intrinsic limit makes the direct growth of high-continuity Ag films with ultimate thinness extremely challenging.

In this work, we propose and demonstrate an ion-beam-based process to fabricate high-quality Ag films with an ultimate thinness down to 4.5 nm. In the process, highly continuous Ag films are first deposited using ion beam sputtering deposition and then thinned back to its thinness limit with angular ion beam polishing. The resultant UTAFs are atomically smooth, demonstrating high figure of merit (FoM) value due to the suppression of surface-roughness-related electrical and optical loss. Especially, the smooth UTAFs show an extremely low transmission optical haze (46 ppm). Simulations present that the ion-beam-sputtering-induced shallow implantation of Ag atoms into the substrates enables the high stability of the UTAFs at the thinness limit. By implementing the uniqueness of shallow implantation in this process, the obtained UTAFs on polymer substrates demonstrate excellent flexibility for applications in flexible optoelectronic devices. The advantages of this process for UTAFs with ultimate thinness, the best possible figure of merit, the lowest haze and excellent flexibility make it a perfect solution to push Ag films as transparent conductors for practical applications.

## Results and discussion
### Fabrication of UTAF

The three-dimensional (3D) flow chart in Fig. 1a schematically demonstrates the thinning-back process for UTAF fabrication. First, a titanium oxide (TiOx) layer is coated on substrates before an Ag film is deposited (Fig. 1aii), which serves as the dielectric seed layer to enhance the wettability of the deposited Ag film. To obtain full and uniform coverage of the seed layer on substrates, an atomic layer deposition technique is used here. Then, a thicker Ag film is deposited by ion beam sputtering (IBS). At the beginning stage of IBS deposition, many isolated Ag clusters are obtained on the seeded substrates due to incomplete wetting (Fig. 1aiii). Via proceeding with the deposition, a sufficiently continuous Ag film is deposited for the subsequent thinning back (Fig. 1aiv) when the deposited thickness is over the percolation threshold. Subsequently, the deposited film is thinned back via angular ion beam polishing (Fig. 1av). Eventually, a UTAF with a desired thickness is achieved (Fig. 1avi). To obtain an excellent UTAF, both high-quality deposition and thinning are key procedures in our developed process. Due to the mismatch of surface energy, Ag adatoms on substrates tend to form tiny dots as nucleation sites (Fig. 1bi) at the beginning of the deposition, even promoted by TiOx seed layer. However, enabled by the additional kinetic energy of the deposited atoms in the IBS process, surface diffusion of Ag adatoms promotes the lateral growth of dots. With the continuous deposition, the tiny dots gradually grow up to islands, coalesce into nanodroplets (Fig. 1bii), merge

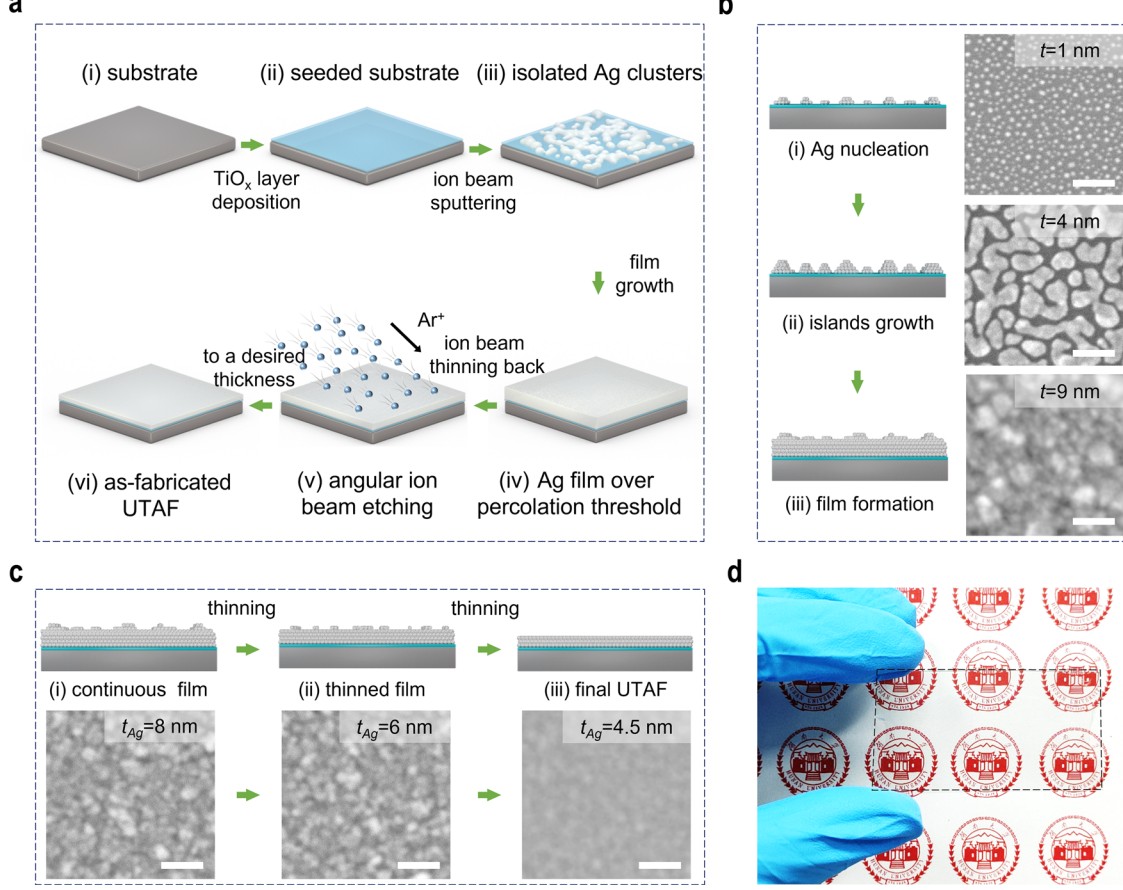

**Fig. 1 | UTAF fabrication. a** 3D schematics showing the ion-beam thinning-back process flow in UTAF fabrication. **b** Series of schematic diagrams exhibiting the detailed growth process of Ag film using the IBS technique. The inset SEM images from top to bottom present the morphologies at nominal thicknesses of 1, 4, and 9 nm. **c** Serial schemes showing the thinning-back and polishing process of the Ag film using glancing ion beam etching with a given incident angle referring to sample surface. The inset SEM images from left to right demonstrate the topography evolution of the Ag film with reduced thicknesses of 8, 6 and 4.5 nm. **d** Photograph of 4.5-nm-thick UTAF on a PET plate (60 mm × 35 mm). Scale bars are 50 nm for all insets in panels (**b**, **c**).

to connected channels, further form a porous film, and eventually achieve a continuous and dense film with a small thickness (Fig. 1biii). To ensure the process more reliable, the thickness of deposited Ag should be 1–2 nm thicker than the percolation threshold. Note that the percolation threshold can be tuned by the wettability of Ag with changing different wetting layers. For example, the percolation threshold for a $TiO_x$-seeded UTAF can decrease from 12 to 9 nm (Supplementary Fig. 1) compared to that of the unseeded UTAF. Based on the as-prepared Ag film (Fig. 1ci), after the subsequent angular etching, the thickness of the Ag film can be reduced to 8, 6, and 4.5 nm. When the film is further thinned to be 3 nm, its morphology becomes discontinuous and changes to be particle shape (Supplementary Fig. 2), though it is still electrically conductive. Moreover, the film surface can be polished by selective planarization of the topographic peaks (Fig. 1cii). With the polishing, the surface roughness is improved (Supplementary Fig. 3). Eventually, a continuous Ag film can be fabricated with a desired thinness and a smooth surface (Fig. 1ciii). To verify the advantages of IBS, we also conducted the thinning-back process based on the magnetron-sputtered and thermally evaporated Ag films. The thinnest achievable thickness of continuous Ag for magnetron sputtering and thermal evaporation were larger than 5.5 and 10.5 nm, respectively (Supplementary Fig. 4), indicating that the IBS technique for Ag deposition is the preferable method to achieve the ultimate thinness. As a proof of concept, a 4.5-nm-thick Ag film is fabricated on a polyethylene terephthalate (PET) plate using this method, and this thin Ag film exhibits high transparency to the naked eye, as shown in Fig. 1d.

## Morphological analysis of UTAF

Based on this process, a continuous Ag film can be thinned back down to 4.5 nm, which is the thinnest one until now. The detailed morphology of the film is investigated, as shown in Fig. 2. In Fig. 2a, long-range continuity of the Ag film without voids is observed in scanning electron microscopy (SEM) images. In contrast, disconnected morphology is observed in the directly deposited film (i.e., without the thinning-back process) with the same thickness (Supplementary Fig. 5). Atomic force microscopy (AFM) analysis further shows the roughness $R_q$ is as low as 0.18 nm, as shown in Fig. 2b. No pit defect is probed using AFM mapping. Cross-sectional transmission electron microscopy (TEM) is used to gain insight into the internal structure of the UTAF. As shown in Fig. 2c, the thickness of seeded Ag film is $4.5 \pm 0.1$ nm. The uniform greyscale in the Ag region depicts no internal porous defects, and the absence of lattice fringes means that an amorphous film is obtained, which benefits the realization of a smooth surface due to the mitigated anisotropic etching of nanocrystals with different facets. The geometrical thickness measured by TEM is also consistent with the optical thickness (4.6 nm) determined by spectroscopic ellipsometry with angstrom resolution, which confirms the long-range uniformity of the UTAF over a large area. The thickness variation at the 1 Å scale indicates the atomical smoothness of the film. To explore the cause of the stability of such thin Ag film, the $Ag/TiO_x$ interface analysis is performed. In Fig. 2d, e, energy-dispersive X-ray spectrometry (EDS) in the scanning TEM mode is applied to probe the element distribution. Line-scan mapping (Fig. 2d, e) suggests the occurrence of interface intermixing between Ag and $TiO_x$, i.e., a part of Ag atoms penetrate into the $TiO_x$

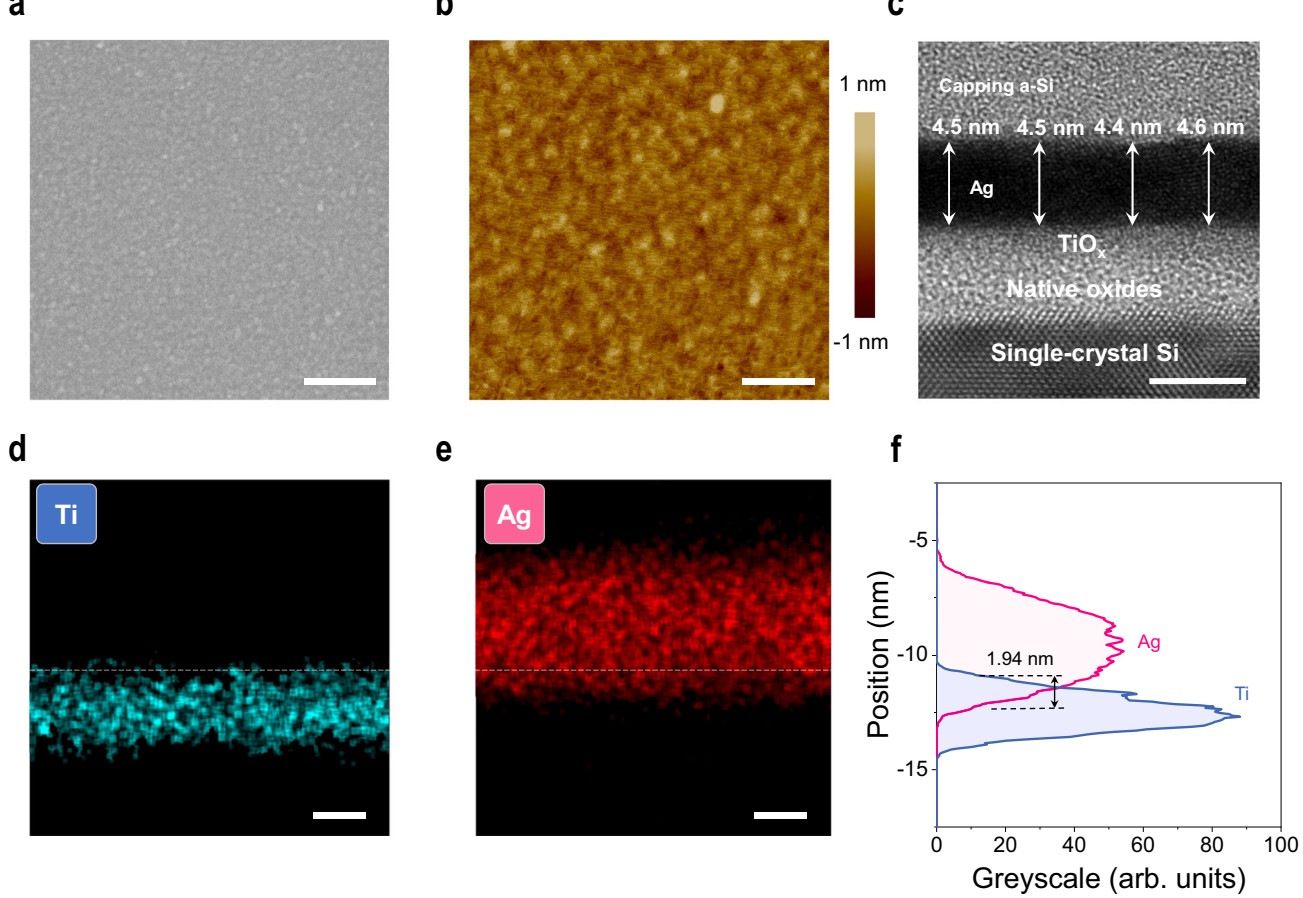

**Fig. 2 | Morphological analysis of a 4.5-nm-thick UTAF. a** Electron micrograph of the surface morphology. Scale bar: 200 nm. **b** Corresponding AFM topography mapping with a scanning resolution of 512 × 512 pixels in a 1 × 1 μm² area. Scale bar: 200 nm. **c** Cross-sectional TEM image of a 4.5-nm-thick UTAF. Scale bar: 5 nm. **d**, **e** EDS mappings of Ti and Ag elements in the scanning TEM mode, respectively. Scale bars: 2 nm. **f** Signal analysis for Ag and Ti distributions at the interface. The signals are extracted from the greyscale integration of corresponding EDS mapping images in panels **d** and **e**.

seed layer. Through the statistics of Ag and Ti signals, an intermixing depth of 1.94 nm (Fig. 2f) was measured, which indicates the generation of Ag implantation in the early stage of Ag film deposition.

## Implantation effect

To further understand the implantation effect in the fabrication of UTAF, systematic simulations were executed. In the stage of IBS deposition, the energy of Ag atoms ejected from the target is dispersive. These Ag atoms with changing energy projecting into the $TiO_x$ underlayer would form an implantation layer on the top of the interface (Fig. 3a inset). Through the Stopping and Range of Ions in Matter (SRIM)[29] simulations, the maximal energy of Ag atoms ejected by 500-eV $Ar^+$ bombardment is determined to be ~19.8 eV. Figure 3a shows that

the average implantation depth for the maximal energy of Ag atoms is 8 Å. The energy dispersion result in a continuous implantation region at the interface, which can serve as a buffer layer to mitigate the aggregation of Ag adatoms induced by the cohesive energy. Even so, the challenge of realizing long-range continuity of UTAF with direct deposition process using the IBS technique remains. In the early stage of IBS deposition, the discontinuity of the Ag implanted layer cannot provide sufficient wettability to expand Ag adatoms into a continuous film because the number of the implanted Ag atoms is not sufficient (top inset in Fig. 3b). Enabled by thinning-back process, a continuous implantation layer is formed when the Ag film over the percolation threshold is deposited using IBS deposition. Empowered by the continuous implantation layer, the 'pinning effect' can suppress the

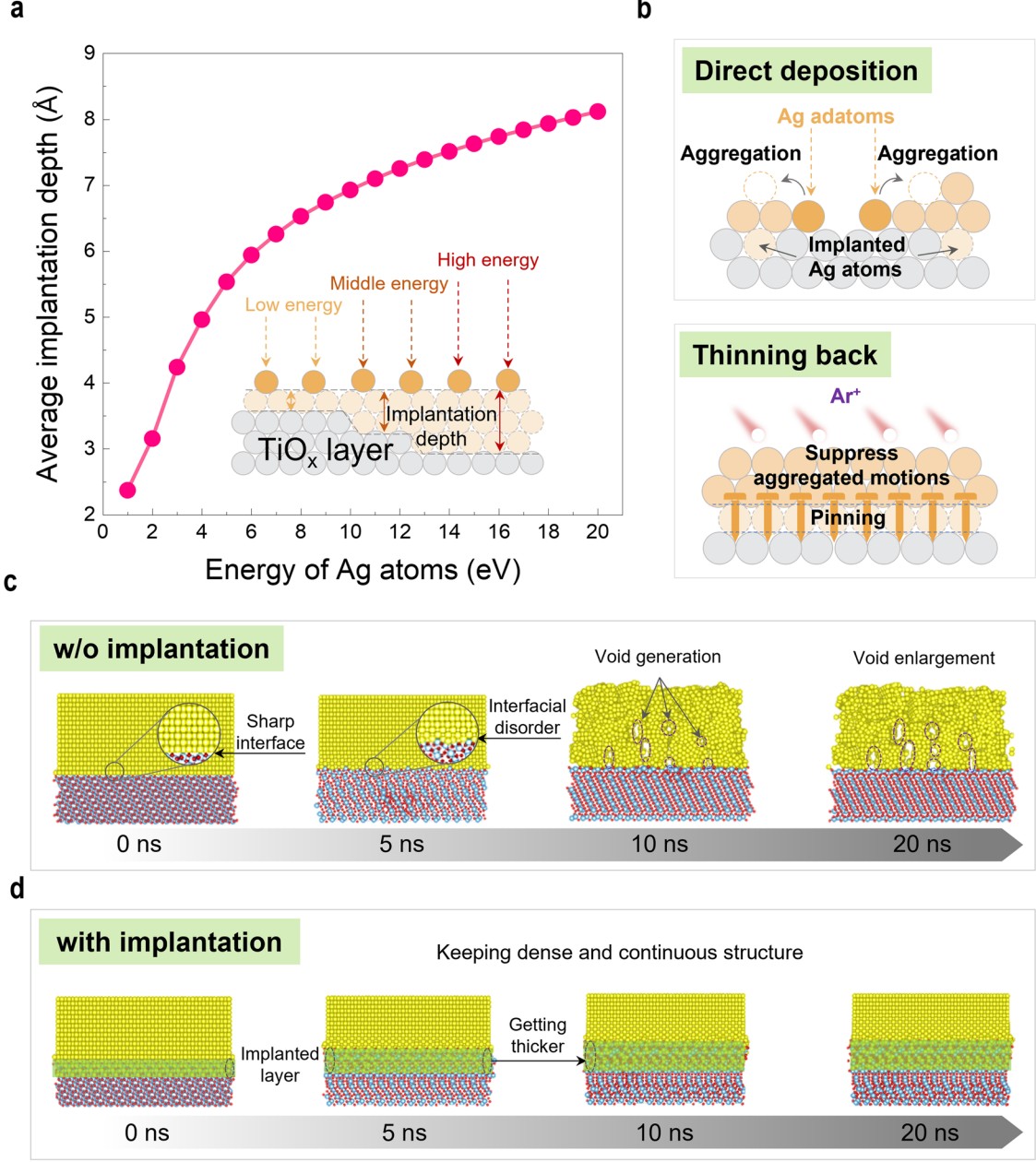

**Fig. 3 | Investigating the effect of Ag implantation. a** Calculated average implantation depth of ejected Ag atoms into $TiO_x$ underlayer as a function of the kinetic energy in SRIM software. The kinetic energy of the Ag atom is in the range of 1–20 eV with an increment of 1 eV. The average implantation depth is the mean value of the vertical projection range from $10^6$ Ag atoms in the single calculation.

Inset scheme demonstrates Ag implantation with various kinetic energies during the IBS deposition. **b** Schematic showing the different behavior of Ag adatoms in UTAF fabrication using direct deposition and thinning-back processes. **c**, **d** MD simulation showing the structural evolution of the ordered-arrangement Ag matrix without and with interface implantation at 300 K, respectively.

dewetting behavior caused by disordered thermal motions of Ag atoms when the UTAF is thinned back to 4.5 nm (bottom inset in Fig. 3b). Such 'pinning effect' is crucial for achieving long-range continuity in UTAFs. Molecular dynamics (MD) simulations further confirm the 'pinning effect' on UTAFs, as shown in Fig. 3c and d. In the absence of implantation (Fig. 3c), the part of the Ag matrix close to the interface becomes disordered at the start of the MD simulation (5 ns). Then, the disordered arrangement of Ag atoms spreads to the whole matrix, and the continuous matrix changes to grain-like structures with voids generation (10 ns). With the proceeding of the disordered thermal motions at 300 K, voids in the matrix are enlarged, finally forming porous defects (20 ns). In the presence of implantation (labeled by green areas, Fig. 3d), the part of the ordered Ag matrix lying on TiOx close to the interface merely appears to change in arrangement, and the implanted layer becomes large in depth. The Ag matrix remains dense and continuous during the whole process of the simulation, which confirms that the "pinning effect" induced by implantation can stabilize UTAFs at the ultimate thinness.

## Optical and electrical characteristics

The superior long-range continuity and excellent surface roughness of UTAFs obtained enable high performance in terms of transparency and electrical conductivity. Here, the visible transmittance and sheet resistance of a seeded UTAF with a reduced thickness during thinning back are measured, as shown in Fig. 4a, b. For a thin metal film, a higher resistivity for thinner films is mainly ascribed to electron scattering

from the surface. The process can be explained by the Fuchs–Sondheimer (FS)[30,31] and Mayadas–Shatzkes (MS)[32,33] models. Here, a comparison of $R_s$ values of UTAFs and fitting data based on the FS-MS model[34] is shown in Fig. 4a. All of the UTAF samples exhibit overall superiority to the equivalents in the FS–MS model. Moreover, the difference between UTAFs and the FS–MS model becomes larger with decreasing Ag thickness ($t_{Ag}$), which is consistent with the change in surface roughness for thinned UTAFs in Supplementary Fig. 3.

The measured transmittance for thinned UTAFs exhibits a coincident downward trend in the visible range and becomes progressively lower at longer wavelengths with increasing $t_{Ag}$ (Fig. 4b), which confirms the sufficient continuity of UTAFs down to $t_{Ag}$ = 4.5 nm. The lower visible transmittance at longer wavelengths makes UTAFs a gray-blue color, similar to low-emission glass. In contrast, the Ag film directly deposited by IBS presents a transmittance dip at approximately $\lambda$ = 548 nm, which is attributed to localized surface plasmon resonance of discontinuous metallic islands[35]. The distinctive spectral behavior of the dashed line verifies the infeasibility of using direct deposition to reduce a continuous Ag film down to 4.5 nm. To provide a direct evaluation of the transparency, the average transmittance ($T_{avg}$) integrated into the visible range is introduced. For $t_{Ag}$ = 4.5 nm, $T_{avg}$ can reach 82%, which is greater than that of commercial ITO ($T_{avg}$ = 77%)[36]. $T_{avg}$ for the discontinuous 4.5-nm-thick (nominal value) Ag film produced via direct IBS deposition is just 63%. As the UTAF thickness increases, $T_{avg}$ drastically decreases to 62% in the case of $t_{Ag}$ = 12.5 nm (Supplementary Table 1). In addition to the long-range continuity, the

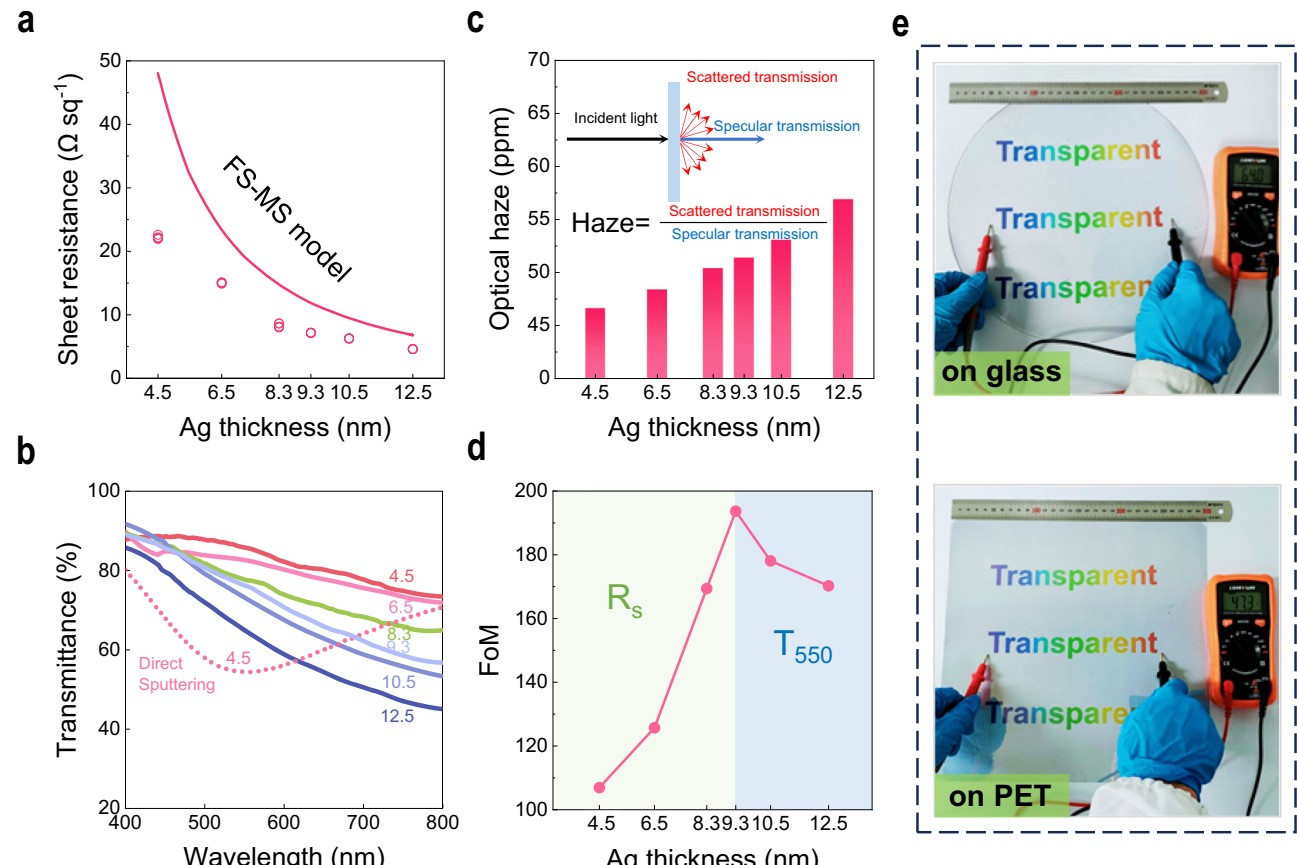

**Fig. 4 | Optical and electrical characteristics of UTAFs. a** $R_s$ for UTAFs with changing reduced thicknesses during thinning back. Raw data (hollow dots) of $R_s$ from five random positions are extracted for each sample. The line fit is the plot of $R_s$ of Ag films in the FS-MS model. **b** Corresponding measured visible transmittance of UTAFs. Solid lines represent continuous UTAFs. The $t_{Ag}$ values labeled on solid lines is the optical thickness measured by ellipsometry. The dashed line represents the particle-shaped UTAF fabricated by direct deposition using the IBS technique. **c** Average visible haze of the UTAFs with different thickness. The inset shows the principle and calculation of optical haze for a film. **d** FoM of UTAFs with changing reduced thickness $t_{Ag}$. **e** Photographs of 9-nm-thick UTAFs on a 12-inch glass wafer (top panel) and a 300 × 300 mm² PET plate (bottom panel).

atomic-scale roughness further suppresses the optical loss caused by scattering and realizes extremely low transmission haze. As shown in Fig. 4c, the integrated haze in the visible range is less than 60 ppm for all UTAF samples, and its value gradually decreased due to the improved smoothness as the Ag thickness $t_{Ag}$ is reduced by the thinning-back step. X-ray diffraction characterization indicates that the improved smoothness is also related to the amorphous nature of the Ag film with thinner thickness (Supplementary Fig. 6). The extremely low optical haze promotes the collimation of transmissive light, which greatly improves the efficacy of light-emitting devices. To assess the overall performance of UTAFs as a transparent conductor, FoM is commonly used as a quantitative index[37]. Figure 4d plots the FoM of the obtained UTAFs. With increasing $t_{Ag}$, the FoM has an optimal value (194) in the case of $t_{Ag} = 9$ nm. Applying the two derived variables $R_s(t_{Ag})$ and $T_{550}(t_{Ag})$ in the FoM function[38] $188.5/(R_s \times (T_{550}^{-1/2}-1))$, we find that the increase of FoM is predominated by the decrease of $R_s$ in the range of 4.5 nm$\leq t_{Ag} \leq 9$ nm (green area). For $t_{Ag} \geq 9$ nm (blue area), the severe deterioration in $T_{550}$ dominantly affects the down slope of the FoM. As a demonstration of this process on uniform large-area fabrication, UTAFs fabricated on a glass wafer (12-inch) and a polymer plate (300 × 300 mm²) exhibit the potential to be TCF candidates (Fig. 4e). The low sheet resistance (7.7 Ω sq⁻¹ on glass; 7.6 Ω sq⁻¹ on PET) illustrates the high continuity of the demonstrated UTAFs. The underlying full color 'Transparent' words present high fidelity in hues, which indicates high transparency in the visible range. Furthermore, the

aging test of UTAF is executed as well. Lasting for a week, the stable transparency and electrical conductivity indicate that UTAFs have the potential to be a promising candidate of ITO alternatives (Supplementary Fig. 7, Table 2).

## Mechanical tests

Contributed from the ductility of metals, UTAFs on polymers are promising candidates for flexible transparent conductors. Here, their flexibility is systemically evaluated. Using a desktop unit for flexibility testing (inset in Fig. 5a), a robust mechanical tensile strength is exhibited by folding a 9-nm-thick UTAF on PET to different curvature radii (R). For the minimal radius R = 3.5 mm, $R_s$ merely increases by 5% (Fig. 5a). For the commercial flexible ITO, $R_s$ drastically increases to 155 times of the original value. In addition to the static strength test, the dynamic fatigue lifetime further demonstrates their flexibility for practical applications. In the cyclic folding test, $R_s$ for UTAF only has a slight increase of 6.58% after $10^5$ cycles. For commercial ITO, the sheet resistance is 601 times larger than the original value after $10^4$ cycles. Morphology characterization clarifies that the drastic degradation of electrical conductance for ITO is ascribed to the presence of massive through cracks (Supplementary Fig. 8). In contrast, UTAF keep a continuous surface during the whole test. The excellent mechanical flexibility permits UTAFs as a promising ITO alternative for flexible optoelectronic devices. Here, we demonstrate a flexible resistive touch panel made of two parallel UTAFs ($t_{Ag} = 9$ nm) on PET with an air gap.

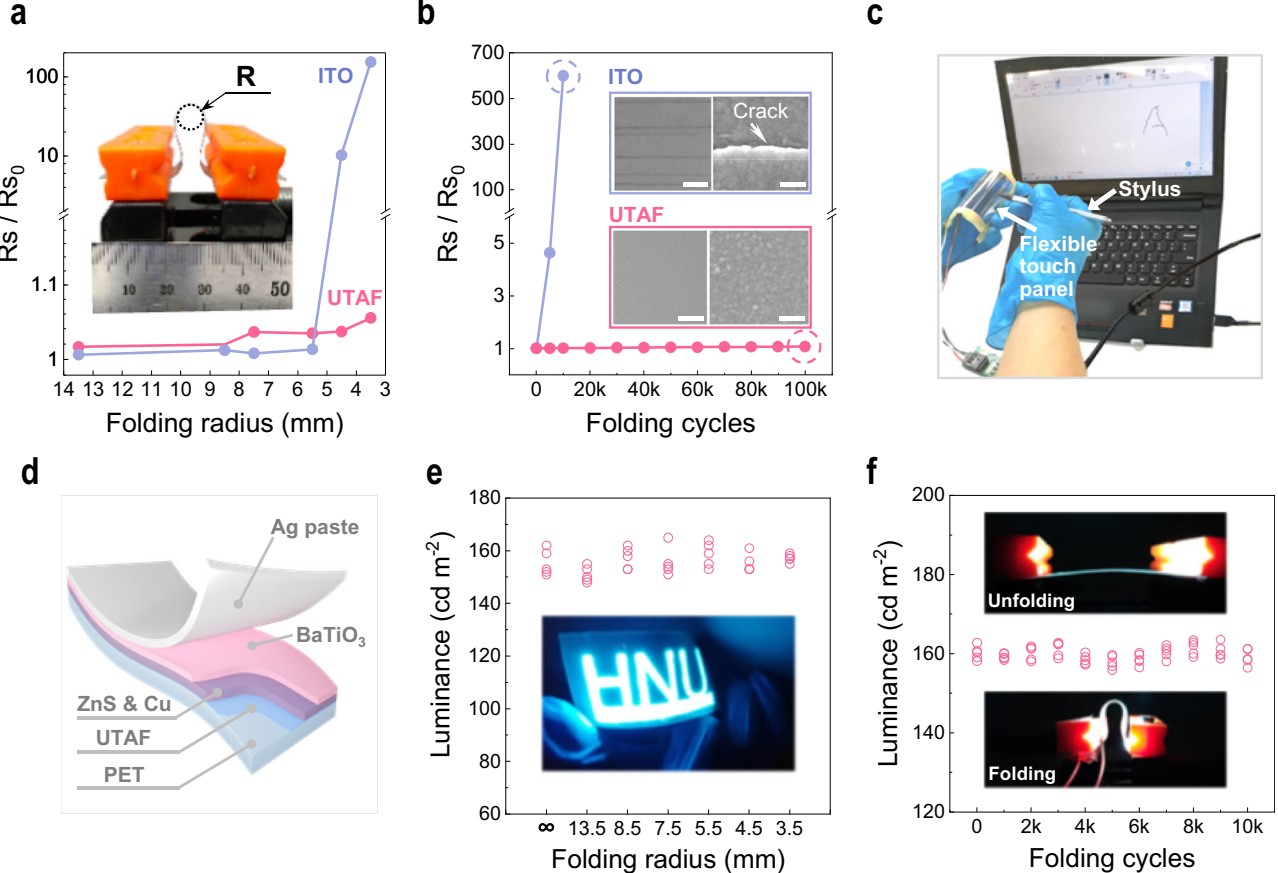

**Fig. 5 | Mechanical flexibility and flexible light-emitting device application of UTAFs. a** Static flexibility of a 9-nm-thick UTAF with different folding radii. The inset photograph presents of the flexibility test unit. **b** Folding stability of the electrical conductance of a 9-nm-thick UTAF and commercial ITO within $10^5$ cycles. Insets show the SEM images and photographs of the ITO and UTAF after $10^4$ cycles and $10^5$ cycles, respectively. Scale bars in insets: 25 μm in left photographs; 100 nm in right SEM images. **c** Proof-of-concept demonstration of a foldable resistive touch panel. **d** 3D scheme for the configuration of a foldable ACEL device. **e** Luminance plot of a foldable ACEL device with changing folding curvature. The ACEL device is driven by a sinusoidal signal (160 V, 16 kHz). The inset picture shows a folded ACEL device with an HNU logo. **f** Luminance stability of the ACEL device undergoing a cyclic folding test. Inset images present the unfolding and folding states for the working ACEL device.

As shown in Fig. 5c, the letter "A" can be clearly written on the folded touch panel.

## Flexible light-emitting devices

With the excellent mechanical flexibility and high transmittance, UTAFs can serve as flexible transparent electrodes for foldable light-emitting devices. In Fig. 5d–f, we demonstrate a foldable alternating current electroluminescent (ACEL) device application using flexible transparent electrodes based on a 9-nm-thick UTAF on PET. Figure 5d presents the configuration of a foldable ACEL device. The flexible ACEL device consists of multiple layers in the following order: a UTAF on PET (bottom electrode), zinc sulfur (ZnS) mixed with copper (Cu) nanoparticles (active layer, ~40 μm), barium titanate ($BaTiO_3$, dielectric layer, ~20 μm), and a top Ag electrode (Ag paste, ~20 μm). When subjected to the folding test, the foldable ACEL maintains a luminance of $160 \, cd \, m^{-2}$ when it is folded with changing curvature radius in static conditions, as shown in Fig. 5e. The electroluminescence (EL) stability of the foldable ACEL device is exhibited by a cyclic folding test (Fig. 5f). After $10^4$ folding cycles, the EL intensity remains at $160 \, cd \, m^{-2}$. For the foldable device, the UTAF provides excellent mechanical flexibility as a uniform carrier collector to support a stable EL intensity over a large area.

In summary, we have developed an ion-beam-based thinning-back strategy to push the thinness limit of a continuous Ag film down to 4.5 nm. Simulations indicate that the shallow implantation capability of the ion-beam-sputtered atoms and the pinning-effect of the implanted atoms are responsible for the improved stability of the UTAFs. With this thinning-back strategy, the final UTAFs exhibit long-range continuity and atomic-level roughness, enabling enhanced transparency and ultimate transmission haze. When combined with polymer substrates, the UTAFs demonstrate robust mechanical stability as a flexible transparent conductor, which boosts the steady EL performance of foldable light-emitting devices. Suitable capping layer can be used to mitigate the oxidization and simultaneously serve as an antireflection (Supplementary Fig. 9, Table 3) or functional device film[8,19]. Appreciating its ultimate capability, we believe that this ion-beam-based thinning-back strategy has the potential to be a general approach capable of producing large-area UTAFs for flexible optoelectronics. Meanwhile, this thinning-back strategy could be extended to fabricate other ultrathin functional films, such as ultra-smooth Au films for low-loss plasmonics, the multilayer reflector for short-wavelength optics, and superconductive films in quantum science.

## Methods

### UTAF fabrication

The fabrication of UTAFs was performed in an IBS system (AS4, Jizhixin Corp.) equipped with dual ion sources. The large-area UTAFs (in Fig. 4e) were fabricated by an industrial-grade IBS system (IBD-XPUT-TER, IBDTEC Inc.). The base pressure of the system was pumped to $5\times10^{-4}$ Pa, and the working pressure was stabilized at $3.2 \times 10^{-2}$ Pa. High-purity argon gas (≥99.999% (5 N)) was ionized for sputtering and etching in dual ion sources. In the sputtering process, the flow rate of Ar gas was 7 sccm for the main ion source, and the voltage of 500 V was applied on the screen grid for accelerating $Ar^+$ ions generated in the main ion sources, and the neutralizer current was set to be 63 mA. Finally, the beam current for ion beam sputtering was about 50 mA. The average growth rate was the thickness of 20 nm for Ag film divided by the corresponding time in accumulation. The resultant deposition rate was set to be $2 \, Å \, s^{-1}$. Thinning the deposited Ag film was in situ executed by the auxiliary ion source in the same vacuum chamber. The incident angle of the ion beam was 45° with reference to the surface plane of the sample. The working pressure was maintained at $3.2 \times 10^{-2}$ Pa. The flow rate of Ar gas was 7 sccm for the auxiliary ion source.

During the thinning-back process, a voltage of 200 V on the screen grid and a beam current of 40 mA were applied for mild and precise etching. The neutralizer current was 48 mA. The etching rate was kept at the level of $1 \, Å \, s^{-1}$.

### Morphology characterization

Field-emission SEM (SIGMA-HD, Carl-Zeiss) and AFM (Dimension Icon, Bruker) were performed to characterize the continuity and roughness of the as-fabricated UTAFs, respectively. To obtain high-quality electronic micrographs, the SEM imaging settings were optimized at an accelerating voltage of 10 kV and a working distance of 3 mm. The roughness information was extracted with high-frequency spatial scanning of $512 \times 512$ pixels in a $1 \, \mu m^2$ area based on tapping mode.

### TEM cross-section sampling and metrology

TEM metrology was utilized to acquire the accurate thickness of the UTAFs. Cross-section sampling of the UTAFs was performed by a focused ion beam system (Helios 5 CX, Thermo Scientific). For a silicon-capped UTAF (4.5-nm-thick), the cross-section sample of the stacked film was sliced to a thickness of 100 nm using a gallium ion beam at 30 kV. Subsequently, a micromanipulator with nanometric accuracy was used to transfer the cross-section sample onto a copper micromesh and fixed by welding using FIB-induced deposition. A 300-kV TEM instrument equipped with a three-lens condenser system and a spherical aberration corrector (Thermis Z, Thermo Scientific) was used for the thickness metrology of the UTAF. EDS mapping was performed in STEM mode.

### Optical transmission measurement in the visible region

A UV-vis-NIR spectrometer (UV-3600i Plus, Shimadzu) was used to measure the transmittance and haze of UTAFs. The visible transmittance of UTAFs was measured while excluding that of the substrates. A four-point probe (M3, HF-Kejing Co., Ltd) was used to measure the sheet resistance of UTAFs with various thicknesses. For wafer-level samples, the whole area of the wafer was divided into twenty-five test fields on the samples, and the sheet resistance data were extracted as the statistical average value from five different positions in each field. Static and dynamic bending for testing the tensile and fatigue mechanical limits was performed with a desktop crank-slide mechanism driven by a stepper electromotor.

### Fabrication of flexible light-emitting devices

First, a 9-nm UTAF on PET served as a transparent bottom electrode. The flexible luminescent active layer was prepared by mixing ZnS:Cu phosphors and resin glue at a weight ratio of 1:1.5. The luminescent layer was sprayed on the flexible transparent electrode. Then the luminescent layer was baked at 90 °C for 30 min. The dielectric layer was prepared by mixing $BaTiO_3$ powder and resin glue at a weight ratio of 1:1 and then sprayed on the luminescent layer. The top electrode was sprayed using Ag paste in the same way.

### Simulations

The injection depth of Ag atoms was simulated by the Transport of Ions in Matter (TRIM)[39] calculation module in SRIM 2013 software. In the simulation, the energy range and the incident angle of Ag atoms was set as 0–20 eV and 0°, respectively. The Ti:O atomic ratio and the density of $TiO_x$ layer was set to 1:2 and $3.32 \, g \, cm^{-3}$ (ref. 40). The energy range of Ag atoms was obtained by a Surface Sputtering calculation in TRIM. In the Surface Sputtering calculation, the energy and the incident angle of $Ar^+$ was set as 500 eV and 45°, respectively.

The MD calculation was carried out in Lammps simulator. The atomic interaction for the Ag−Ti−O system was described by the eam/alloy potential supported by J.B. Adams and X.Y. Liu[41]. Referring to the

data of Ag abundance in $TiO_x$ layer, we set the fraction of Ag atom in the $TiO_x$ layer to be 23%. With a conjugate gradient algorithm, the initial stable structure of the $Ag/TiO_2$ model was obtained via energy minimization. Then the generated atomic configuration was equilibrated at room temperature for 300 ps under a Nose–Hoover thermostat (NPT ensemble). The stress in all three dimensions was controlled at approximately zero after relaxation. Each sample was then equilibrated under the NPT ensemble at a constant temperature of 300 K to achieve an equilibrium state with zero pressure for 10 ns. Subsequently, the system temperature was reduced from 400 K–300 K for annealing for 10 ns under the NPT ensemble (under 1 atmosphere). Next, MD simulations were further carried out for 20 ns with a time step of 1 fs per integration step under the NVT ensemble conditions (300 K).

## Data availability

The data that support the findings of this study have been included in the main text and Supplementary Information. All other relevant data supporting the findings of this study are available from the corresponding authors upon request.

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

## Acknowledgments

H.D. and Y.C. acknowledges the support by the National Key Research and Development Program of China (2022YFB4602600). H.D. acknowledges the support by the National Natural Science Foundation of China (52221001). M.Z. acknowledges the support by the National Natural Science Foundation of China (12104182). The authors also thank Dr. Peng Dai for the analytical calculations of the film transmittance and helpful discussions.

## Author contributions

H.D. and Y.C. conceived this study. D.M., M.J., H.Y., Q.W., and M.Z. performed the experiments and measurements in this study. F.F. drew the 3D schematic illustrations. B.F. carried out the simulations. All authors participated in the manuscript review.

## Competing interests

The authors declare no competing interests.
