## [Peer Review File · Nature Communications]

Pushing the thinness limit of silver films for flexible optoelectronic devices via ion-beam thinning-back processREVIEWER COMMENTS

Reviewer #1 (Remarks to the Author):

Pursuing the best possible transparent conductive electrodes (TCO) is of great importance for improving the performance of modern optoelectronic devices. Ag film is supposed to be an ideal candidate, but its performance is limited by the Ostwald ripening when the thickness is less than 10 nm. In this work, the authors reported that the thickness limit of continuous Ag film could be reduced to 4.5 nm through a two-step ion beam processing technique. The achievement is ascribed to two key points. On the one hand, angular ion beam etching is used for thinning and polishing the initially deposited Ag film. On the other hand, high-energy Ag atoms are implanted into the underlying TiO_x layer, which plays an important role in the stabilization of Ag film at the extreme thinness as it undergoes ion beam bombardment. The authors utilize these two strengths to enhance the quality of UTAF, resulting in films with an atomically smooth surface. The performance tests verify that the ultrasmooth surface benefits high electrical conductivity and visible transparency. Moreover, the ability of high-quality UTAF fabrication on polymers demonstrates the potential for flexible transparent conductors in flexible devices. Mechanical flexibility tests indicate a fatigue lifetime of 10⁵ cycles and excellent robustness of large-curvature folding. This novel fabrication approach could potentially offer a universal solution for producing ultrathin and ultrasmooth functional films. I recommend the publication of this work in Nature Communications. Before publication, some issues should be clarified or discussed in the manuscript.

- (1) Although Ag is a suitable material for transparent conductors at extremely small thicknesses, the atmospheric stability of UTAF is crucial for replacing ITO or not. The authors should provide some practical and available pathways to mitigate the oxidation of UTAF-based transparent conductors for practical applications.
- (2) Is this approach reported in this work available for other noble metals? For example, ultrathin Au films. If possible, can the authors provide such kind of results?
- (3) The authors claim that the implantation during the IBS procedure plays a key role in achieving such thin Ag film because the pinning effect from implantation can stabilize UTAF and avoid the occurrence of dewetting behavior. Can the authors compare the results of UTAF fabrication using evaporation or magnetron sputtering?
- (4) Is it possible to further reduce the thickness of the UTAF beyond 4.5 nm? What is the ultimate limit?
- (5) To clearly demonstrate the priority of UTAF, the authors should thoroughly reviewed the progress of the ITO alternatives, also provide a table of figures for comparison in the morphology and properties, especially in the optoelectronic properties and FoM.

Reviewer #2 (Remarks to the Author):

This is a very solid work on a new method of producing ultra-thin Ag film with record 4.5nm thickness that maintains a smooth morphology, coupled with comprehensive characterization and simulation. The approach is drastically different from conventional methods, achieved by ion beam sputtering of a pre-deposited Ag film. The "implanted" Ag atoms server as pinning centers that stabilize the ultra thin Ag

layer during the ion etching process. The clarification of the commonly used FOM with film thickness and the relative importance of resistance and transparency reduction is another notable contribution. The novelty of the process and the impressive results warrant its acceptance by Nat. Comm after a revision. I'd like to list the following points for authors to clarify in the revision:

1. It is said that "the prefabricated film is thinned back via angular ion beam polishing", please provide some details on the process, which will benefit the readers. Presumably the angular variation of ion beam is important. Can such a process be simulated using the similar method authors used to study the possible Ag "implantation" process?

2. Please comment on the stability of the 4.5nm Ag film when taken out of the IBS chamber. How long can the observed effect last in ambient environment. It will be great to supplement stability data at elevated temperature, but this can be in a follow-up work.

3. Fig. 3 shows ejected Ag atom range: is this calculation based on Ag diffusion in Ag layer or in TiO₂? To correspond to the actual situation, both should be considered. Also since the initial Ag layer is ~10nm thick, presumably the ejected Ag energy of ~19eV is for Ag atom at the bottom of the Ag layer? Could authors estimate the average lateral spacing of Ag atoms being pushed into the TiO₂ layer

Response to Reviewers

Reviewer #1: *Pursuing the best possible transparent conductive electrodes (TCO) is of great importance for improving the performance of modern optoelectronic devices. Ag film is supposed to be an ideal candidate, but its performance is limited by the Ostwald ripening when the thickness is less than 10 nm. In this work, the authors reported that the thickness limit of continuous Ag film could be reduced to 4.5 nm through a two-step ion beam processing technique. The achievement is ascribed to two key points. On the one hand, angular ion beam etching is used for thinning and polishing the initially deposited Ag film. On the other hand, high-energy Ag atoms are implanted into the underlying TiO_x layer, which plays an important role in the stabilization of Ag film at the extreme thinness as it undergoes ion beam bombardment. The authors utilize these two strengths to enhance the quality of UTAF, resulting in films with an atomically smooth surface. The performance tests verify that the ultrasmooth surface benefits high electrical conductivity and visible transparency. Moreover, the ability of high-quality UTAF fabrication on polymers demonstrates the potential for flexible transparent conductors in flexible devices. Mechanical flexibility tests indicate a fatigue lifetime of 10⁵ cycles and excellent robustness of large-curvature folding. This novel fabrication approach could potentially offer a universal solution for producing ultrathin and ultrasmooth functional films. I recommend the publication of this work in Nature Communications. Before publication, some issues should be clarified or discussed in the manuscript.*

Response: Thanks for the positive comments from the reviewer.

Comment #1: *Although Ag is a suitable material for transparent conductors at minimal thicknesses, the atmospheric stability of UTAF is crucial for replacing ITO. The authors should provide some practical and available pathways to mitigate the oxidation of UTAF-based transparent conductors for practical applications.*

Response: Thanks for this helpful suggestion. As the reviewer mentioned, the oxidation of UTAF is a vital point of chemical stability at ambient conditions as an ITO alternative. To improve the stability of UTAF, we proposed two feasible antioxidation methods for two different kinds of device applications as follows.

The first kind of devices is based on the interaction of the physics fields (e.g., electromagnetic, electrical, magnetic, and thermal fields) with UTAF or applying the physics field to drive the device. In these devices, a transparent coating for physics field propagation can be deposited on UTAF, and the coating should be chemical inertia in oxidation as well. For example, Parylene, an optically transparent and dielectric material, could be used to prevent the UTAF from oxidation in wet or air atmospheres, because it is hydrophobic and dense to insulate the penetration of water and oxygen. Some oxides (e.g., SiO₂, TiO₂) and nitrides (e.g., Si₃N₄, AlN) are effective materials for antioxidation as well because they are conductive for electromagnetic, electrical, and magnetic fields. We can further deposit oxide or nitride layers on UTAFs using ion-beam sputtering, and these top antioxidation layers also play the role of antireflection coating for enhancing optical transparency.

The second kind of devices needs UTAF to play the role of transparent conductor for electron transportation to drive devices. The antioxidation coatings not only have good optical transmittance in the visible range but also possess unique electrical conductivity. Moreover, matching the work function in the semiconductor optoelectronics further limits the material options for antioxidation coating. For example, the UTAF as a transparent electrode needs a transparent conductive capping layer (e.g., ZnO, MoO₃, AZO, and ZnS) [Yun J. *et al.*, *Adv. Funct. Mater.*, **2017**, *27*, 1606641; Schubert S. *et al.*, *Adv. Funct. Mater.*, **2012**, *22*, 4993-4999; Zhang Y. *et al.*, *Mat. Sci. Semicon. Proc.*, **2023**, *165*, 107643; Leng J. *et al.*, *J. Appl. Phys.*, **2010**, *108*, 073109] for antioxidation, and we also consider the work function matching of the capping layer with the upper layer in the configuration of OLED and photovoltaic cells for the efficiency of devices as high as possible [Kang H. *et al.*, *Nat. Commun.*, **2015**, *6*, 6503; Zhang C. *et al.*, *Adv. Opt. Mater.* **2021**, *9*, 3, 2001298; Sergeant P. *et al.*, *Adv. Mater.* **2012**, *24*, 728-732]. To provide the referable suggestion about UTAF antioxidation to readers, we have added the statement “Suitable capping layer can be used to mitigate the oxidization and simultaneously serve as an antireflection (**Supplementary Fig. 9, Table 3**) or functional device film” to the revised manuscript (Page 16 in the revised manuscript).

Comment #2: *Is this approach reported in this work available for other noble metals? For example, ultrathin Au films. If possible, can the authors provide such kind of results?*

Response: Thanks for this constructive suggestion. Following the reviewer’s suggestion, we demonstrated the fabricated ultrathin Au film using ion beam thinning back. As shown in **Figure R1**, a continuous Au film is thinned to 7 nm in thickness. Here, we used the native oxide on a silicon wafer as a wetting underlayer, the 7-nm thick Au film presents greatly high continuity in a long range and extremely low RMS in surface roughness. To enable the readers to know the material expansibility of the ion-beam thinning-back process, we have revised “this thinning-back strategy could also be extended to fabricate other ultrathin functional films such as ultrathin gold films and superconductive films for low-loss plasmonics, short-wavelength optics and quantum materials applications.” to be “this thinning-back strategy could be extended to fabricate other ultrathin functional films such as ultrasmooth Au films for low-loss plasmonics, the multilayer reflector for short-wavelength optics, and superconductive films in quantum science.” in the main text (Page 16 in the revised manuscript) and added the results into the *Supplementary Information* as Supplementary Fig. 10.

Figure R1. Fabricated 7-nm thick Au film by ion beam thinning-back process. SEM morphology (a) and AFM topography (b) of ultrathin Au film on a single-polishing silicon substrate. The surface roughness is 0.155 nm RMS in statistics. The scanning resolution in AFM topography mapping is 512×512 pixels in 1×1 μm^2 . The thickness is the optical equivalent value which is determined by spectroscopic ellipsometry. Scale bars: 100 nm.

Comment #3: *The authors claim that the implantation during the IBS procedure plays a key role in achieving such thin Ag film because the pinning effect from implantation can stabilize UTAF and avoid the occurrence of dewetting behavior. Can the authors compare the results of UTAF fabrication using evaporation or magnetron sputtering?*

Response: Thanks for this professional suggestion. Following the reviewer’s suggestion, we compared the thinned-back UTAFs based on the thermally evaporated and magnetron-sputtered Ag

films. The Ag films are set to have the same initial thickness which is larger than the percolation threshold. Subsequently, the Ag films were thinned till the change of continuity in morphology. As shown in **Figure R2**, the evaporated Ag film presents porous morphology when its thickness reduces to $t_{Ag} = 10.5$ nm. The magnetron-sputtered Ag film presents the continuity transition for $t_{Ag} = 5.5$ nm. This comparison among deposition methods verifies two advantages of ion-beam thinning back based on a dual ion source sputtering system. (1) In our work, the process portfolio combining ion beam sputtering with the thinning-back step is performed in a vacuum chamber with single loading. For evaporation and magneto sputtering, exchanging samples between different fabrication vacuum chambers is required, which might lead to unwanted oxidation and contaminations. (2) Higher energy of ejected Ag atoms by ion-beam sputtering enables larger-range surface diffusion of adatoms, which makes the deposited film dense. The denser deposited film is beneficial to obtain the smaller thinness limit during thinning back.

To provide a helpful suggestion about the ion-beam thinning-back process to readers, we have inserted “To verify the advantages of IBS, we also conducted the thinning-back process based on the magnetron-sputtered and thermally-evaporated Ag films. The thinnest achievable thickness of Ag for magnetron sputtering and thermal evaporation were larger than 5.5 and 10.5 nm, respectively (**Supplementary Fig. 4**), indicating that the IBS technique for Ag deposition is the preferable method to achieve the ultimate thinness.” in the revised manuscript (Page 6 in the revised manuscript).

Figure R2. SEM images of thinned-back UTAfs based on the deposited continuous Ag films by magnetron sputtering (a) and evaporation (b). The initial thickness of deposited Ag films is set to be 15 nm, and the thinned-back thickness is a nominal value based on the etching rate of ion-beam polishing. Scale bars: 200 nm.

Comment #4: *Is it possible to further reduce the thickness of the UTAF beyond 4.5 nm? What is the ultimate limit?*

Response: Thanks for this professional comment. To respond the reviewer's suggestion, we performed ion-beam polishing to further thin 4.5-nm-thick Ag film on the TiO_x wetting layer/SiO₂ substrate. As shown in **Figure R3**, we found that the morphology transition from high continuity to porous when its thickness reduces to 3 nm (nominal value). To claim the thinness limit in this work, we have inserted the statement "the thickness of the Ag film can be reduced to 8 nm, 6 nm, and 4.5 nm. When the film is further thinned to be 3 nm, its morphology becomes discontinuous and changes to be particle shape (**Supplementary Fig. 2**), though it is still electrically conductive." into the main text (Page 5 in the revised manuscript) and added Supplementary Fig. 2 in **Supplementary Information** section.

Figure R3. Morphology of 3-nm-thick UTAF. The result is obtained by further thinning the 4.5-nm-thick UTAF sample. Scale bar: 200 nm.

However, the claimed thinness limit of 4.5 nm could be further improved by optimizing the process. Two pathways listed as follows could be considered. First, the underlayer with higher wettability can be applied. Compared to TiO_x, ZnO is a wetting material with better wettability, which could decrease the percolation threshold of Ag film growth [Yun J. *et al.*, *Advanced Functional Materials*, **2017**, *27*, 1606641] and thus can help to realize a smaller thinness limit of UTAF. Second, the low-temperature process would be the other pathway to improve the thinness limit of UTAF. The lower process temperature can suppress the dewetting of Ag atoms [Lemasters R. *et al.*, *ACS Photonics*, **2019**, *6*, 2600-2606]. In the reported work, the temperature of the substrate

was kept at 15°C via a water-cooling system. A lower temperature using cooling media such as liquid nitrogen may be helpful to further reduce the thinness limit.

Comment #5: *To clearly demonstrate the priority of UTAF, the authors should thoroughly reviewed the progress of the ITO alternatives, also provide a table of figures for comparison in the morphology and properties, especially in the optoelectronic properties and FoM.*

Response: Thanks for this professional comment. Following the reviewer’s suggestion, we evaluated the performance of UTAFs fabricated by our method and other processes in terms of surface roughness (**Table R1**), sheet resistance R_s (**Table R2**), and transmittance at the wavelength of 550 nm T_{550} (**Table R3**). High quality in roughness and continuity enables the UTAF to possess high transparency and electrical conduction with the suppression of photon and electron scattering. Enabled by *in-situ* subtractive processing from the auxiliary ion source, not only the thickness of the deposited film is reduced, but also the surface roughness is mitigated. Furthermore, pure Ag material in UTAF configuration further improves the electrical and optical performance. As shown in **Table R3**, even though thinner Ag film can be applied, the addition of lossy materials (e.g., Al, Cu) as a wetting layer degrades the optical transparency of UTAFs.

Table R1. Surface roughness compared with previous studies.

Ag film thickness	RMS (nm)	Process	Reference
9 nm	0.859	Al-doped Ag	Zhang C. et al. , Adv. Mater. , 2014 , 26, 5696-5701
9 nm	0.84	Polyethyleneimine wetting	Yang X. et al. , Sci. Rep. , 2017 , 7, 44576
9 nm	0.23	Polyethyleneimine wetting	Kang H. et al. , Nat. Commun. , 2015 , 6, 6503
9 nm bare Ag	2.61	Direct thermal evaporation	Kang H. et al. , Nat. Commun. , 2015 , 6, 6503
9.3 nm	0.2	Ion beam thinning back	Our work (reducing from 13 to 9.3 nm)

Table R2. Sheet resistance compared with previous studies.

Ag film thickness	R_s (Ω sq ⁻¹)	Process	Reference
9 nm	19	Al-doped Ag	Zhang C. et al. , Adv. Mater. , 2014 , 26, 5696-5701
9 nm	6.9	Polyethyleneimine wetting	Yang X. et al. , Sci. Rep. , 2017 , 7, 44576
9 nm	9	Polyethyleneimine wetting	Kang H. et al. , Nat. Commun. , 2015 , 6, 6503

9 nm bare Ag	N/A	Direct thermal evaporation	Kang H. et al. , Nat. Commun. , 2015 , 6, 6503
9.3 nm	7.16	Ion beam thinning back	Our work

Table R3. Transmittance compared with previous studies.

Ag film thickness	T_{550} (%)	Process	Reference
6 nm	80	Al-doped Ag	Zhang C. et al. , Adv. Mater. , 2014 , 26, 5696-5701
6 nm	80	Cu-doped Ag	Huang J. et al. , Sol. Energ. Mat. Sol. C. , 2018 , 184, 73-81
6.5 nm	83	Ion beam thinning back	Our work

Compared with other transparent electrode techniques (e.g., graphene and Ag nanowires) (**Table R4**), the transmittance of UTAF is slightly lower, but it can be easily solved by introducing an antireflection coating to form constructive interference. **Figure R4** shows the antireflection result of UTAF based on the ZnO/Ag/ZnO (ZAZ) structure. T_{550} mapping of ZAZ structure with a fixed t_{Ag} (8 nm) and variable t_{ZnO} at the top and bottom layer from 0 to 80 nm is calculated. The maximal T_{550} is 96.8% when the top and bottom t_{ZnO} are 61 and 59 nm, respectively. **Figure R4b** shows the calculated and experimental visible transmittance of 61 nm ZnO/8 nm Ag/59 nm ZnO in the visible region. As highlighted in **Table R4**, the ZAZ structure has a high transmittance (96.8%@550 nm), good conductivity ($10.16 \Omega \text{ sq}^{-1}$), and demonstrated a high overall performance with the quantitative assessment of FoM value. The FoM value of 1132 for ZAZ is much higher than those of transparent electrodes prepared by other techniques. However, due to the use of ZnO, the transmittance at shorter wavelengths is relatively low, which might be caused by the slightly higher absorption of ZnO from the oxygen vacancies during ion-beam sputtering [Dostanko A. P. *et al.*, *Semiconductors*, **2014**, 48, 9, 1242-1247].

Because this work focuses on the fabrication of pure Ag film, we tend not to compare it with other configurations in the main text of the manuscript to avoid possible confusion. A systematical comparison study could be conducted via additional work in the future.

Figure R4 The design of antireflection for UTAF based on ZAZ structure. (a) Calculated T_{550} mapping of ZAZ as a function of top and bottom t_{ZnO} variables. Ag thickness t_{Ag} is fixed to be 8 nm. (b) The calculated spectrum and experimental spectra of ZAZ structure with setting thicknesses of 61 nm top t_{ZnO} /8 nm t_{Ag} /59 nm bottom t_{ZnO} .

Table R4. The performance of ZAZ compared with other electrodes

Techniques	$T_{550 \text{ nm}}$ (%)	R_s ($\Omega \text{ sq}^{-1}$)	FoM	Reference
ITO	85.7	13.36	176	Lim, J. W. et al. , Opt. Express , 2014 , 22, 26891-26899
AgNWs	92	32	138	Ge Y. et al. , J. Am. Chem. Soc. , 2018 , 140, 193-199
Graphene	97	125	98	Bae, S. et al. , Nat. Nanotechnol. , 2010 , 5, 574-578
CNT	74	22	53	Guo C. F et al. , Mater. Today , 2015 , 18, 143-154
Ag grid	90	75	46	Guo C. F et al. , Mater. Today , 2015 , 18, 143-154
ZnO/Ag/ZnO (61/8/59 nm)	96.8	10.16	1132	Our work

Note: $FoM = 188.5 / (R_s \times (T_{550}^{-1/2} - 1))$

Reviewer #2: *This is a very solid work on a new method of producing ultra-thin Ag film with a record 4.5 nm thickness that maintains a smooth morphology, coupled with comprehensive characterization and simulation. The approach is drastically different from conventional methods, achieved by ion beam sputtering of a pre-deposited Ag film. The “implanted” Ag atoms serve as pinning centers that stabilize the ultrathin Ag layer during the ion etching process. The clarification of the commonly used FOM with film thickness and the relative importance of resistance and transparency reduction is another notable contribution. The novelty of the process and the impressive results warrant its acceptance by Nat. Comm after a revision. I’d like to list the following points for authors to clarify in the revision:*

Response: Thanks for the positive comments from reviewer.

Comment #1: *It is said that “the prefabricated film is thinned back via angular ion beam polishing”, please provide some details on the process, which will benefit the readers. Presumably the angular variation of ion beam is important. Can such a process be simulated using the similar method authors used to study the possible Ag “implantation” process?*

Response: Thanks for this professional comment. Following the reviewer’s suggestion, we further provided the process details in the method section of UTAF fabrication (Page 16 in the revised manuscript). We have revised the procedure details of thinning back as follows.

“The fabrication of UTAFs was performed in an IBS system (AS4, Jizhixin Corp.) equipped with dual ion sources. The large-area UTAFs (in **Fig. 4e**) were fabricated by an industrial-grade IBS system (IBD-XPUTTER, IBDTEC Inc.). The base pressure of the system was pumped to 5×10^{-4} Pa, and the working pressure was stabilized at 3.2×10^{-2} Pa. High-purity argon gas ($\geq 99.999\%$ (5 N)) was ionized for sputtering and etching in dual ion sources. In the sputtering process, the flow rate of Ar gas was 7 sccm for the main ion source, and the voltage of 500 V was applied on the screen grid for accelerating Ar^+ ions generated in the main ion sources, and the neutralizer current was set to be 63 mA. Finally, the beam current for ion beam sputtering was about 50 mA. The average growth rate was the thickness of 20 nm for Ag film divided by the corresponding time in accumulation. The resultant deposition rate was set to be 2 \AA s^{-1} . Thinning the deposited Ag film was *in situ* executed by the auxiliary ion source in the same vacuum chamber. The incident angle

of the ion beam was 45° with reference to the surface plane of the sample. The working pressure was maintained at 3.2×10^{-2} Pa. The flow rate of Ar gas was 7 sccm for the auxiliary ion source. During the thinning-back process, a voltage of 200 V on the screen grid and a beam current of 40 mA were applied for mild and precise etching. The neutralizer current was 48 mA. The etching rate was kept at the level of 1 \AA s^{-1} .”

Because polishing a film is achieved in the manner of atom ejection from the surface using a flood ion beam, precise modeling of this fabrication process is multiscale and requires the construction of mesoscopic morphology in the form of an atom matrix. Meanwhile, the interaction among atoms may be heterogeneous and homogeneous, so the interaction strength would be variable for amorphous and crystalline structures and hard to quantify. Meanwhile, it is difficult to visualize the mesoscopic morphology evolution of a deposited film and further evaluate the surface quality of the polished film by varying the incident angle of the ion beam in the SRIM software. Up to now, there is no commercial simulator that can accurately calculate the complicated physics model. However, the control experiments can directly provide the preferable incident angle for optimizing the surface quality of polishing. Previous works investigated the influence of ion beam incident angle on surface roughness and polishing efficiency in experiments [Chkhalo N. *et al.*, *Appl. Optics*, **2016**, 55, 1249-1256; Gosset N. *et al.*, *J. Micromech. Microeng.*, **2015**, 25, 095011; Frost F. *et al.*, *Nucl. Instrum. Meth. B.*, **2004**, 216, 9-19; Glöersen P. G. *et al.*, *J. Vac. Sci. Technol. A.*, **1975**, 12, 28-35]. The investigation clarified that the incident angle is a key parameter to influence surface roughness and etching rate. **Figure R5** shows the dependence of surface roughness and etching rate on the incidence angle of ions for the fused silica polishing in the work [Chkhalo N. *et al.*, *Appl. Optics*, **2016**, 55, 1249-1256]. In **Figure R5a**, obvious deterioration of surface roughness occurs at the incidence of ions in the range of angles of $45\text{-}80^\circ$ (referring to the sample surface), and low surface roughness can be obtained when the incidence angle of ions is below 45° . Meanwhile, the maximum etching rate occurs at the incidence angle of around 45° . Therefore, the incidence angle of 45° is a good point for balancing the surface roughness and processing efficiency during ion beam polishing in this work. However, it's worth noting that the balance point would be variable based on the different materials [Gosset N. *et al.*, *J. Micromech.*

Microeng., **2015**, *25*, 095011; Frost F. *et al.*, *Nucl. Instrum. Meth. B.*, **2004**, *216*, 9-19; Glöersen P. G. *et al.*, *J. Vac. Sci. Technol. A.*, **1975**, *12*, 28-35].

Figure R5. Dependence of surface roughness (a) and etching rate (b) on the incidence angle of ions for the fused silica polishing [Chkhalo N. *et al.*, *Appl. Optics*, **2016**, *55*, 1249-1256]. The angle refers to the sample surface.

Figure R6 schematically shows the surface morphology transition of deposited film using ion beam angular (a) and normal (b) etching. With angular incident (**Figure R6a**), the microscopic “peaks” on the surface are easier to be removed by angular ions bombardment, and the “valleys” are etched slowly due to the shadow shielding from “peaks”, by which the resultant surface gradually be planarized. In contrast, normal etching performed non-selective removal to “peak” and “valley” features. Therefore, angular ion beam etching can generate the effect of thinning and smoothing.

Figure R6. Schematic illustrations of surface morphology of deposited film before and after angular (a, c) and normal (b, d) ion beam etching.

Comment #2: Please comment on the stability of the 4.5 nm Ag film when taken out of the IBS chamber. How long can the observed effect last in ambient environment. It will be great to supplement stability data at elevated temperature, but this can be in a follow-up work.

Response: Thanks for this professional suggestion. Following the reviewer's suggestion, we have executed the stability test of the 4.5 nm UTAF when taken out of the IBS chamber. We monitored the change of average visible transmittance and sheet resistance in the aging test lasting for a week. The average visible transmittance decreased by 0.7%, and the sheet resistance increased with an increment of 10%. Meanwhile, the surface morphology maintains high continuity, as shown in **Figure R7**. To avoid possible confusion to readers, we have inserted "Furthermore, the aging test of UTAF is executed as well. Lasting for a week, the stable transparency and electrical conduction indicate that UTAFs have the potential to be a promising candidate of ITO alternatives (**Supplementary Fig. 7, Table 2**)." in the main text (Page 13 in the revised manuscript).

Table R5. The average visible transmittance and sheet resistance of 4.5-nm thick UTAF when taken out of the IBS chamber in a week.

Day	Average visible transmittance (%)	Sheet resistance ($\Omega \text{ sq}^{-1}$)
1 st day	83.80	18.35±0.68
2 nd day	83.76	18.81±0.76
6 th day	83.62	19.38±1.01
7 th day	83.61	19.87±0.90

Figure R7. The surface morphology of as-fabricated (a) 4.5 nm UTAF and that after 7 days (b) when taken out of the IBS chamber and placed in air atmosphere ($\sim 16^\circ\text{C}$, 40% relative humidity). Scale bars: 200 nm.

As the reviewer suggested, we performed the high-temperature stability of bare UTAF and ZnO-coated UTAF. As shown in **Figure R8**, the 4.5-nm-thick bare UTAF can maintain the continuous morphology at 150°C but change to nanoparticles when it is heated and kept at 200°C for 5 mins. With a top layer coating of ZnO (20 nm), the corresponding UTAF can be stable at 300°C for 5 mins. The result is a rough study about high-temperature stability, and we will further get a deep insight into this investigation in follow-up works.

Figure R8. The morphology evolution of bare (a, c, e) and ZnO-coated (b, d, f) UTAF at high-temperature conditions. The high-temperature test is performed in the nitrogen gas atmosphere for 5 min. Scale bars: 200 nm.

Comment #3: *Fig. 3 shows ejected Ag atom range: is this calculation based on Ag diffusion in Ag layer or in TiO₂? To correspond to the actual situation, both should be considered. Also since the initial Ag layer is ~10 nm thick, presumably the ejected Ag energy of ~19 eV is for Ag atom at the bottom of the Ag layer? Could authors estimate the average lateral spacing of Ag atoms being pushed into the TiO₂ layer.*

Response: Thanks for the professional comment.

Fig. 3 shows that the ejected Ag atoms range is calculated based on energetic Ag atoms projecting in the TiO_x layer using the Monte Carlo program [H. Hofsäss *et al.*, *Appl. Surf. Sci.* **2014**, *310*, 134-141]. The calculation result is not obtained based on the diffusion theory. In SRIM, the projection range of an ion in thickness is its path length (distance of travel) projected onto a

coordinate axis. SRIM calculates the motion trajectory of ions in solids based on the collision processes and an energy loss model during ion interactions with matter. The stopping power (dE/dx) is a key parameter in this context, representing the energy loss per unit path length of the ion as it travels through the material. The ion implantation depth is then determined by integrating the stopping power along the ion trajectory. As the reviewer mentioned, in the actual deposition, ejected Ag atoms projecting into the TiO_x underlayer lead to the change of chemical composition due to Ag implantation, which affects the projection depth in the underlayer. We used SRIM to simulate the depth of ejected Ag atoms into Ag-doped TiO_x (The atom fraction of Ag referring to the measured data in the EDS line-scan mapping), and the calculated depth is about 7.7 angstrom, smaller than that in pure TiO_x . However, the actual penetrating depth of Ag is much larger than the calculated results. The difference in depth between simulation and experiment is ascribed to diffusion and implantation effects. Tsukizoe and coworkers also found that the actual depth of the Ag atom implanted in Cu is greater than that in the calculation, and he attributed this difference in depth to the dislocation and Frenkel point defects [T. Tsukizoe *et al.*, *J. Appl. Phys.* **1977**, *48*, 4770]. Meanwhile, the low-energy ion penetration is applied for the low-temperature silicon oxidation. The oxidation thickness is largely dependent on the surface implantation and radiation-enhanced diffusion process [S. S. Todorov *et al.*, *J. Vac. Sci. Technol. B* **1988**, *6*, 466–469]. The intermixing phenomenon at heterogeneous interfaces in IBS deposition has been also found in the fabrication of Mo/Si EUV multilayer reflectors. The multilayer reflector consists of 40 pairs of ultrathin Mo and Si layers. The IBS deposition has been the mainstream technology for the manufacturing of EUV multilayer reflectors. As shown in **Figure R9**, the interfacial intermixing phenomenon (grey regions) exists in the multilayer structure. The intermixing layer in thickness is 2.19 nm for Mo on Si and 1.16 nm for Si on Mo, respectively. Some previous works reported that the asymmetric intermixing is ascribed to the different diffusion potential of Mo on Si and Si on Mo deposition conditions [A. Köhler *et al.*, *Mat. Res. Soc. Symp. Proc.*, **2003**, 749; M. J. H. Kessels *et al.*, *Nuclear Instruments and Methods in Physics Research B*, **2004**, *222*, 484-490; C. Largeron *et al.*, *Philosophical Magazine*, **2006**, *86*, 19, 2865-2879]. Hence, the diffusion and implantation effect converge to the intermixing phenomenon in actual IBS depositions.

Figure R9. Cross-sectional TEM image of Mo/Si multilayer deposited by IBS technique, and the thickness of Mo and Si layer in single pair is 2.4 nm and 4.2 nm, respectively. The bright and dark part is the Si and Mo layer, respectively. The gray region represents the intermixing layer between the Mo and Si layers.

As commented by the reviewer, 19.8 eV is the maximal energy of ejected Ag atoms onto the substrate and the energy of starting ejected Ag atoms projecting into substrate layers. The following ejected atoms would pass through the initial deposited Ag layer, the heavier Ag layer will decelerate the following Ag atoms and result in the shallower implantation depth. For the energy of 19.8 eV at the starting projection, the average implantation depth is about 0.8 nm. Ascribed to the growth of Ag layer on TiO_x , the implantation depth becomes shallower for following Ag atoms as the deposition process proceeds. Finally, the implanted layer is continuous and generates the pinning effect to stabilize the UTAF during the ion-beam thinning back procedure.

Here, we estimated the average lateral spacing of Ag atoms being pushed into the TiO_x layer with the calculation of Ag abundance penetrating the TiO_x underlayer. The integrated average of the Ag atom fraction in the implantation range (position interval of the FWHM of Ti peak) is about 23%. Referring to the Ag abundance in the TiO_x layer, we set the atom fraction (23%) of the Ag atom penetrating the TiO_x layer in the MD simulation. The diagram of EDS line-scan mapping for Ag and Ti normal to its interface is shown in **Figure R10**. Meanwhile, we have updated the MD

simulation part in *Methods* section with the description of the relevant setting “Referring to the data of Ag abundance in TiO_x layer, we set the fraction of Ag atom in the TiO_x layer to be 23%.”.

Figure R10. The Ag and Ti distribution analysis on the normal line of UTAF on the TiO_x wetting layer. The line-scan mapping is performed in XTEM.

REVIEWERS' COMMENTS

Reviewer #1 (Remarks to the Author):

This manuscript has been well modified and properly responded. I satisfy with this version and recommend to acceptance in Nature Communications.

Reviewer #2 (Remarks to the Author):

I am satisfied with authors' detailed response and further analysis with additional experiments.